

# Vertical Distribution of Heat and Sodium Fluxes in the Mesopause Region Measured by Sodium Lidar Over Hainan, China (109°E,19°N)

Xingjin Wang[1,2,3], Xin Fang[1,2,3*], Wenhao Gao[1,3,†], Xianhang Chen[1,3],Tai Liu[4],
Chengyun Yang[1,3], Tingdi Chen[1,2,3], Tao Li[1,3], Xianghui Xue[1,2,3]

[1]School of Earth and Space Sciences, University of Science and Technology of China, Hefei, China.

[2]Hefei National Laboratory, University of Science and Technology of China, Hefei, China.

[3]CAS Key Laboratory of Geospace Environment, School of Earth and Space Sciences, University of Science and Technology of China, Hefei, China.

[4]Department of Geophysics, College of the Geology Engineering and Geomatics, Chang'an University, Xi'an, 710054, China.

[†]deceased, 9 September 2024.

[*]*Correspondence to*: Xin Fang (xinf@ustc.edu.cn)

**Abstract.** We present the first lidar-based characterization of seasonal variations in gravity-wave–induced vertical heat flux, sodium flux, and associated parameters—sodium density and temperature—between 80 and 100 km over Hainan, China (19° N, 109° E). Observations were carried out using a narrow band sodium lidar equipped with a laser frequency-locking and real-time monitoring module, achieving a root-mean-square (RMS) frequency stability of 0.5 MHz. Since February 2024, the system has provided continuous measurements of mesospheric sodium density, temperature, and vertical wind. The lidar results are generally consistent with coincident satellite measurements and model simulations at the near geographic location. Observations indicate that the highest temperatures below 95 km occur in May and November, with seasonal patterns closely matching from the SABER satellite data. The annual mean vertical heat flux shows two peak descent rates, −0.25 K m$^{-1}$ s$^{-1}$ at 83 km and −0.80 K m$^{-1}$ s$^{-1}$ at 89 km, corresponding to a cooling rate of approximately 40 K day$^{-1}$ between 82 and 97 km. The sodium flux reveals two pronounced maxima exceeding −50 m s$^{-1}$ cm$^{-3}$ at 88 and 90 km, with the resulting dynamical transport producing a maximum net sodium loss of 165 cm$^{-3}$ h$^{-1}$ near 91 km. These findings provide direct evidence that the gravity-wave-breaking occurrences modulate both the thermal structure and chemical composition of the mesopause range.





## 1. Introduction

In the mesopause region, ranging from approximately 80 to 100 km in altitude, temperature and wind are critical atmospheric parameters for understanding the dynamics of this region. Measurements of atmospheric parameters in this region enable investigations of dynamic and photochemical processes in the upper mesosphere, which also serves as a transition zone of importance to aviation and aerospace activities (Sheng et al., 2025). Since the 1970s, ground-based instruments and spaceborne sensors have

been extensively employed to measure key parameters (Cox et al., 1993; Gardner et al., 1986; She et al., 1998; Vincent and Reid, 1983; Wu et al., 2008). Compared to the satellites and medium-frequency radars, narrow band sodium lidars offer the unique capability to simultaneously measure temperature and horizontal wind in the mesopause region by leveraging the high-resolution sodium emission spectrum (Arnold and She, 2003; She and Yu, 1994; Vincent and Reid, 1983).

Heat and compositional variations in the mesopause range are primarily driven by atmospheric gravity waves (AGWs) through their influence on global circulation. AGW-driven vertical transport of heat and constituents is expressed by the vertical fluxes of sensible heat and species densities, defined as the expected values of the product of vertical wind fluctuations (w′) and the corresponding fluctuations in temperature (T′) or constituent density ($\rho'_{Na}$), induced by the gravity wave spectrum (Chu et al., 2022;

Gardner, 2024; Liu and Gardner, 2005). Measurements from the Sodium Resonance Lidar at the Starfire Optical Range (SOR) in New Mexico show that the maximum downward dynamical flux of sodium (Na) can reach $-280$ m·s$^{-1}$·cm$^{-3}$ at~88 km, indicating that dynamical transport often exceeds the vertical transport associated with eddy diffusion (Liu and Gardner, 2004).

In this study, we report on a newly developed narrow band, high-spectral-resolution sodium lidar system,

designed by the University of Science and Technology of China (USTC). This system builds upon the earlier USTC narrow band sodium temperature--wind lidar and is based on a self-developed pulsed dye amplifier (PDA) system. Specifically, the new 589 nm lidar employs a more stable and powerful single-frequency Raman fiber amplifier, along with new designs that integrate an enhanced timing control system and a more precise frequency locking unit based on modulation transfer spectroscopy (MTS)

technology. Furthermore, beat frequency technology is used to monitor frequency jitter of between pulsed and continuous-wave laser in real time.



From February 2024 to January 2025, routine measurements of sodium density, temperature, and vertical wind were obtained to calculate sensible heat flux and sodium flux over Hainan, China. Section 2 provides a detailed description of the lidar system and its technical improvements. Section 3 presents monthly averaged results of sodium density and temperature, including comparisons with SABER satellite and the MSISE model data, thereby confirming the reliability of the lidar data for scientific research. Section 4 discusses the sensible heat flux associated with AGWs and its implications for heating and cooling effects in the background atmosphere over Hainan. Section 5 summarizes the main conclusions.

## 2. Lidar System Setup

In this section, we present an overview of the technical enhancements implemented in the lidar system. A schematic diagram of the narrow band sodium lidar deployed in Hainan is shown in Fig. 1, a photograph of the system in Fig. 2, and the corresponding system parameters are summarized in Table 1.

The lidar system consists of three main components: a transmitter, a receiver, and a signal detection and control module. The lidar transmitter system includes three commercial lasers (Semiconductor seed laser, Raman amplified laser, 1064 nm seed laser and Nd: YAG laser), a self-made pulsed dye amplifier (PDA), and a set of custom-engineered module for absolute frequency locking with self-calibration. A semiconductor seed laser (DL Pro) emits 1178 nm continuous-wave laser with a maximum power of 70 mW. The 1178nm seed laser after pre-amplification is split into two beams. One beam is frequency-doubled using a Periodically Poled Lithium Niobate (PPLN) crystal for frequency locking, while the other beam is frequency-shifted by three frequencies using a fiber acousto-optic frequency shifter (AOFS) module and then undergoes secondary amplification to obtain a three-frequency 589nm laser, with the average power of approximately 1.5 W.

A Nd: YAG laser (Continuum PLS 9050e) produces 532 nm pulses at 50 Hz with a maximum energy of 600 mJ and a pulse duration of 9 ns. The amplified three-frequency 589 nm laser is further amplified and pulsed through a three-stage traveling-wave amplification system by the 532 nm pulsed laser at 400 mJ and the 589 nm continuous-wave laser at 1.5 W, yield a three-frequency 589 nm pulsed laser with 30 mJ energy, 6 ns pulse width, and a beam diameter of 7 mm.



The backscattered photons after the interaction of the 589 nm pulsed laser with sodium atoms in the atmosphere are collected by a telescope with a diameter of 1 m and 1.7 m focal length. The photons then pass through a narrow band optical filter, a converging lens, and a collimating lens, before being focused onto the active area of a photomultiplier tube (PMT). A high-speed photon counter records the photons counts, from which vertical profiles of sodium atomic density, temperature, and vertical wind velocity in

the sodium layer are derived.

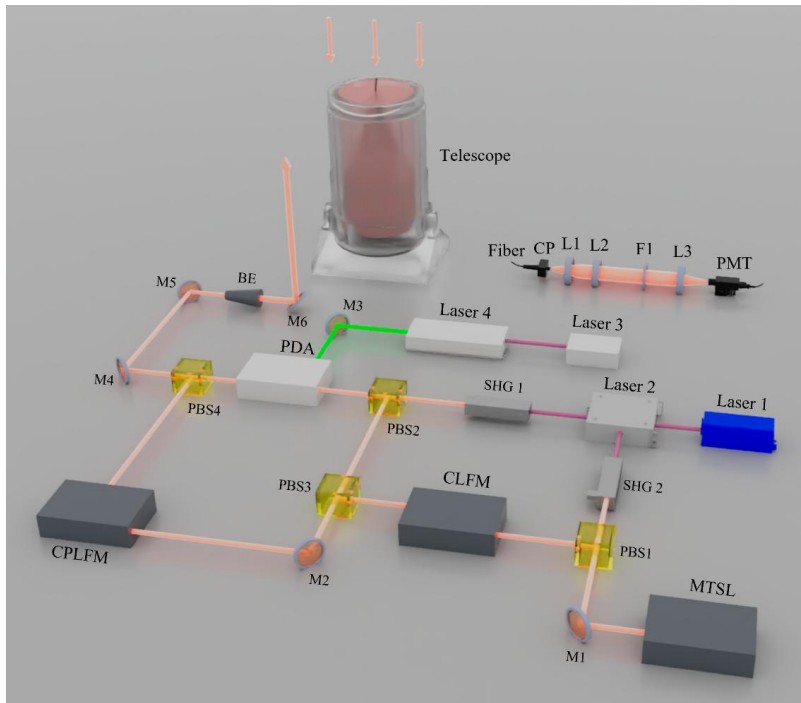

**Figure 1. Schematic diagram of the lidar system. Laser 1, semiconductor seed laser; Laser 2, Raman amplified laser; Laser 3, 1064 nm seed laser; Laser 4, Nd: YAG laser; SHG, second-harmonic generation; PBS, polarizing beam splitter; M, mirror; BE, beam expander; CP, Chopper; L, lens; F, filters; PMT, photo multiplier tube; MTSL, modulation transfer spectroscopy (MTS) locking unit; CLFM, continuous laser frequency monitoring unit; CPLFM, continuous and pulsed laser frequency monitoring unit.**





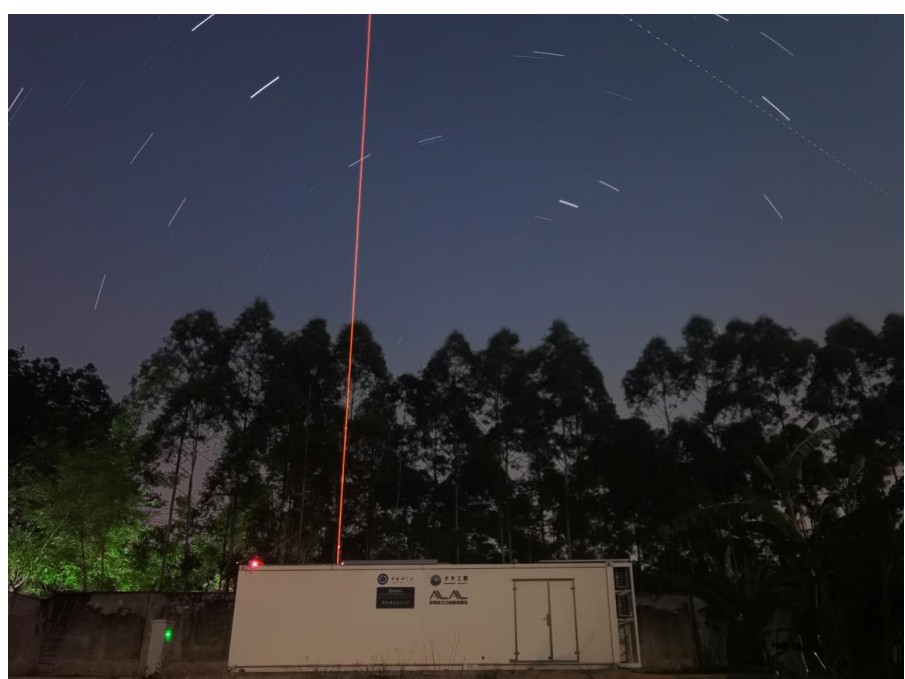

**Figure 2. Picture of the lidar system.**

**Table 1.** Lidar Parameters

| | | |
|---|---|---|
| | Wavelength | 589.158 nm |
| | CW seeder laser into PDA | ＞0.5 W |
| | Nd: YAG laser | 20 W |
| Lidar Transmitter | Linewidth | 120 MHz |
| | Pulse energy of 589 nm | ~30mJ |
| | Repetition rate | 50 Hz |
| | Pulse width | 6 ns |
| | Beam divergence | 0.7 mrad |
| | Telescope aperture | 1 m |
| Lidar Receiver | Field of view | 0.88 mrad |
| | Bandwidth of filter | 1 nm |
| | Bin width | 61.44 m |

The main technical improvements of this lidar system. Fig. 3a illustrates the schematic of the laser emission system. The absolute frequency locking and monitoring module of the laser is shown in detail, comprising a modulation transfer spectroscopy (MTS) unit, a continuous-wave and pulsed 589nm frequency monitoring unit based on iodine absorption, and a laser frequency jitter monitoring unit between the continuous-wave and pulsed 589nm laser.



### 2.1 Modulation Transfer Spectroscopy (MTS) Locking unit

The modulation transfer spectroscopy (MTS) method involves applying a modulation signal to an electro-optic modulator (EOM), which generates sidebands that beat with the carrier frequency. A photodetector is then used to detect and demodulate the resulting heterodyne signal, allowing atomic modulation transfer spectral lines to be resolved.

To eliminate Doppler broadening, a pair of counter-propagating laser beams is implemented. As illustrated in Fig. 3c, the high-power pump beam is modulated by an EOM, while the low-power beam serves as the probe beam. The two beams are spatially overlapped and propagate in opposite directions. When the laser frequency approaches the atomic transition, sidebands are generated on the probe beam via the four-wave mixing effect, transferring the modulation from the pump beam to the probe beam. A photodiode (PD1) detects the probe beam, and demodulation of the signal yields the frequency stabilization error signal.

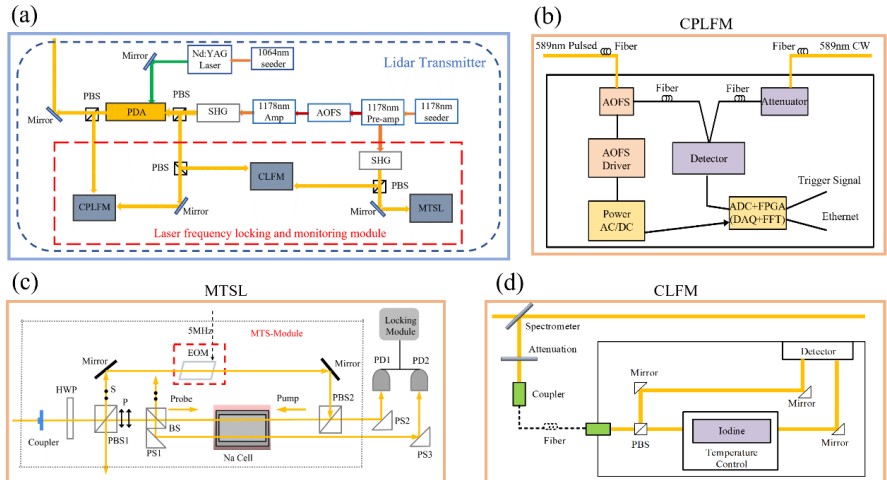

**Figure 3. Schematic diagram of the laser emission part and laser frequency locking and monitoring module system of the lidar system. (a) the laser emission part of the lidar system; (b) continuous and pulsed laser frequency monitoring (CPLFM) unit; (c) modulation transfer spectroscopy locking (MTSL) unit; (d) continuous laser frequency monitoring (CLFM) unit.**

### 2.2 Continuous Laser Frequency Monitoring unit

Because the single-frequency Raman amplifier generates two laser beams, the MTS was used to lock the frequency of one low-power beam to the sodium D2a line. Simultaneously, the frequency difference between the second beam and the locked beam is tracked in real time. The iodine molecular transmission



spectrum near the sodium D2a line is employed to calibrate the relationship between laser frequency and iodine cell transmittance, enabling frequency-difference tracking. Once the laser frequency is locked, the iodine transmittance is monitored in real time to evaluate frequency stability and locking accuracy. The optical configuration is shown in Fig. 3d.

Figure 4 presents the results of laser frequency monitoring. In the unlocked state, the laser frequency drifts freely with an amplitude of ~20 MHz over 22 minutes. After applying MTS-based frequency locking, the frequency jitter is reduced to within ±1 MHz of the designated output frequency.

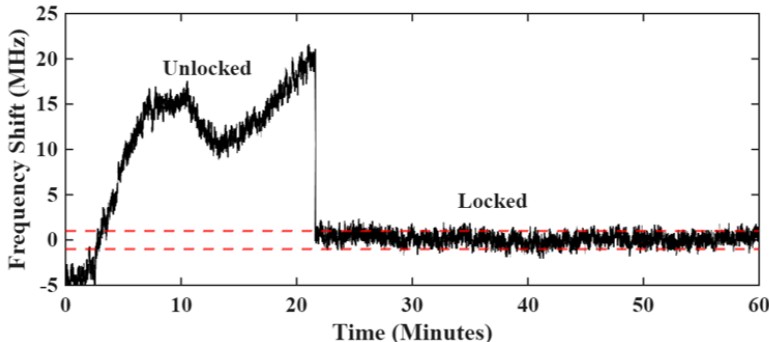

**Figure 4. Continuous laser frequency monitoring results.**

**2.3 Continuous and Pulsed Laser Frequency Monitoring unit**

Since the detected signal originates from pulsed laser with a 50 Hz repetition rate, it is essential to measure the frequency shift induced by the chirp effect of the PDA. To achieve this, three acousto-optic frequency shifters (AOFS) are used to shift the frequency of the frequency-locked 589 nm continuous-wave beam by 1150 MHz. The pulsed laser emerging from the PDA is then mixed with the shifted

continuous-wave laser to generate a heterodyne signal. This signal is detected and demodulated to determine the real-time frequency shift. Subtracting the known 1150 MHz offset yields the actual frequency difference between the pulsed and continuous beams. The optical configuration is illustrated in Fig. 3b.

As shown in Fig. 5b, the frequency difference between the 589 nm continuous-wave and pulsed beams

measured on 7 July 2024, was approximately −14.9 MHz. The monitoring results (Fig. 5a) indicate a frequency stability of ±1 MHz, its root mean square (RMS) is almost 0.5MHz. Thus, the total frequency offset between the pulsed laser used for observations and the continuous-wave laser locked to the sodium D2a line was −15.9 MHz, corresponding to a vertical wind offset of −9 m s$^{-1}$.





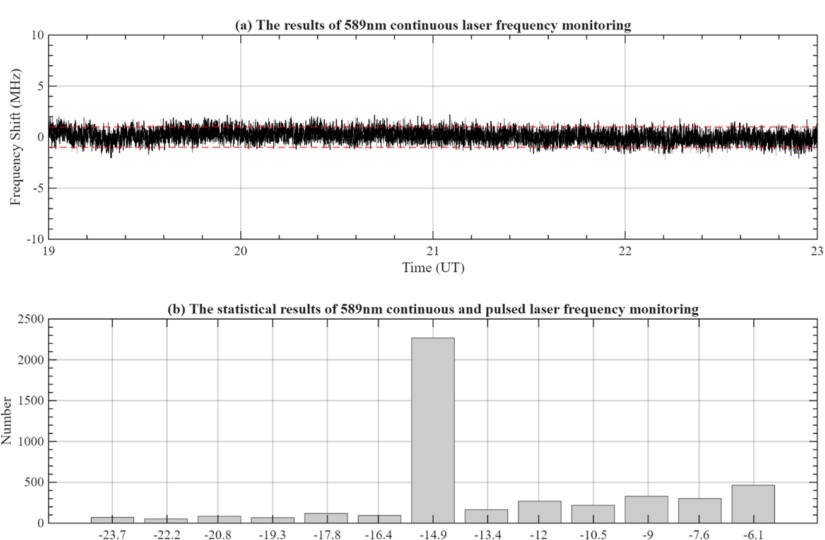

**Figure 5. Results of the laser frequency locking and monitoring module on 7 July 2024. (a) the results from continuous laser frequency monitoring unit; (b) the statistical results from continuous and pulsed laser frequency monitoring unit.**

Figure 6 compares the lidar vertical wind measurements with the nighttime mean wind on 7 July 2024. The average vertical wind velocity throughout the night is $\sim-9$ m s$^{-1}$, consistent with the offset determined from frequency monitoring unit.

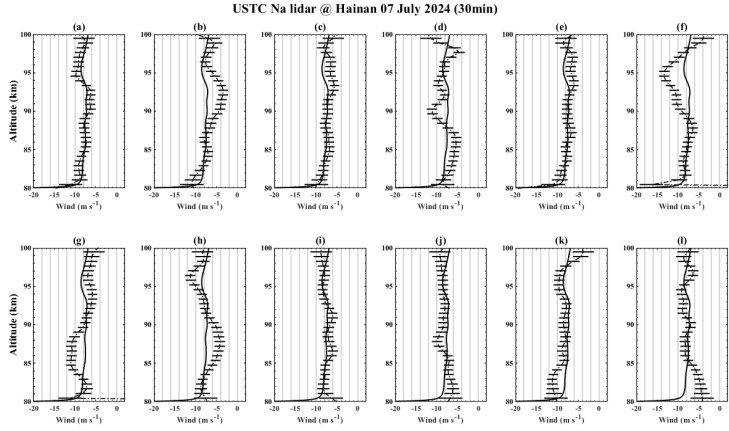

**Figure 6. Sodium layer vertical wind (solid line) and the mean of the nighttime wind (dashed line) at 30 minutes on 7 July 2024. Error bars are shown as thin lines.**



**2.4 Summary of Section 2:**

The integration of absolute frequency locking and real-time self-calibration allows accurate measurement of the frequency difference between the pulsed 589 nm laser and the sodium D2a spectral line. These capabilities ensure precise vertical wind retrievals, validating the stability and performance of the Hainan sodium lidar.

**3. Na Lidar Initial Results and Processing**

**3.1 Initial Results**

The raw signals were recorded as photon counts with a vertical resolution of 61.44 m and a temporal resolution of 1 minute (corresponding to 3000 laser pulses). The detector gate was opened 10 μs after each laser pulse, retaining only the backscattered signal. Sodium density, temperature, and vertical wind were calculated by integrating the photon counts over 30 minutes, with a vertical resolution of 2 km

(Wang et al., 2025).

In this section, we compare sodium layer temperatures measured by the lidar system with those from SABER satellite observations and the MSISE model across different seasons (Fig. 7). More wave-like structures in the SABER and lidar temperature profiles are likely attributable to gravity waves, since SABER requires only ~1.5 min to produce a single profile. Although measurement locations may differ

by several hundred kilometers, the temperature profile trends among the three datasets are broadly consistent, confirming the reliability of the lidar observations.




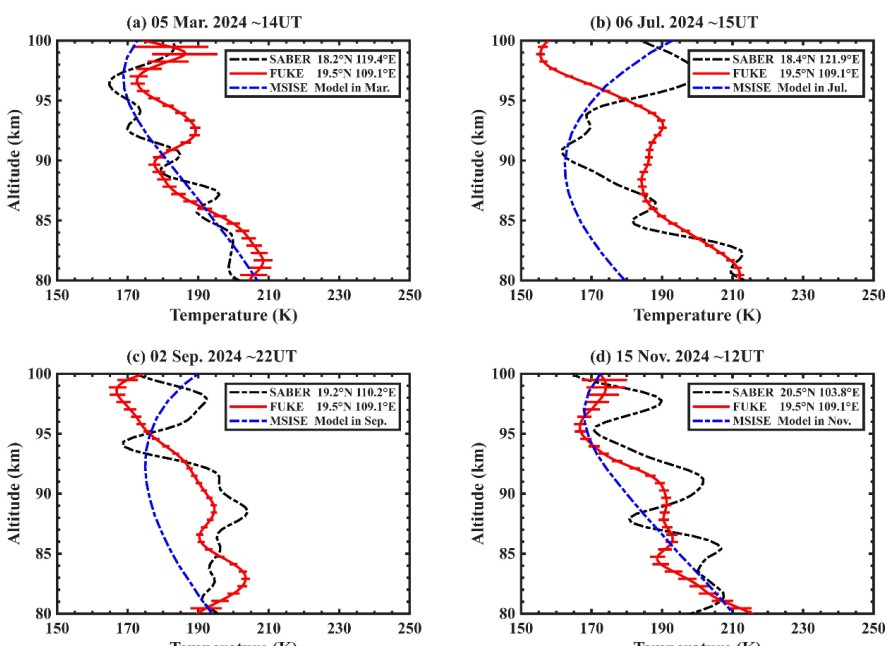

**Figure 7. Typical profiles of temperature measured by lidar (red solid line), SABER (black dashed line) and MSISE (blue dashed line) in different seasons. (a) spring, (b) summer, (c) autumn, (d) winter. Error bars are shown as thin lines.**

## 3.2 Seasonal Variations in Sodium Density and Temperature

To investigate seasonal variations in sodium density and temperature over Hainan, 53 observational samples were selected between February 2024 and January 2025, with a cumulative duration exceeding 326 hours. Figure 8 shows the monthly distribution of the number of nights with valid lidar data, along with the corresponding observation times.




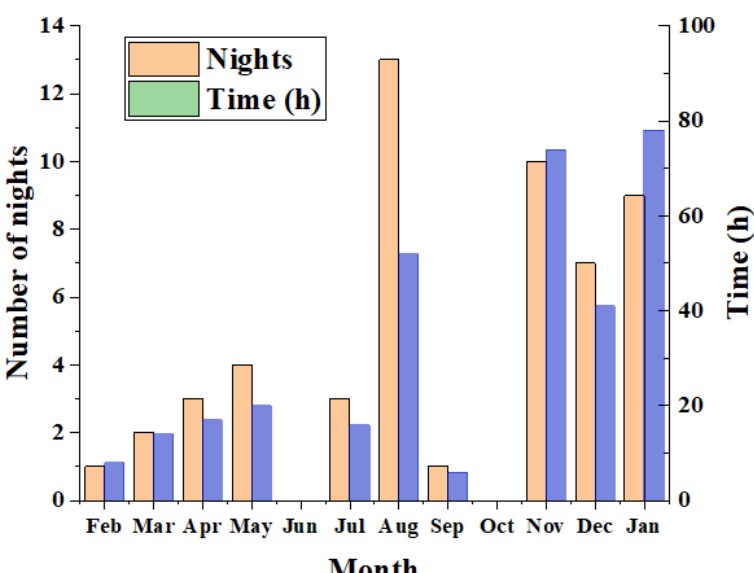

**Figure 8. Histogram of number of nights and hours with valid data observed by the USTC sodium lidar at Hainan.**

As illustrated in Fig. 9, the monthly-averaged sodium number density exhibits a peak altitude range

between ~85 and 96 km, centered near 91 km. The centroid height varies seasonally: ~93–95 km in spring, decreasing to ~90 km in late summer and autumn (beginning in August). In winter (around November), the peak density reaches its annual maximum of over 4200 cm⁻³ at 91 km, whereas from February to September, the peak remains in the range of 3000–3500 cm⁻³.

Overall, seasonal variability in sodium density over Hainan is pronounced, with monthly averages

generally above 3000 cm⁻³. When combined with the monthly-mean temperatures shown in Fig. 10, the relationship between sodium density and temperature is consistent with previous findings from the Hefei narrow band sodium lidar: below 95 km, sodium density correlates positively with temperature, suggesting strong chemical control on sodium production; above 95 km, sodium density exhibits a negative correlation with temperature (Li et al., 2018).



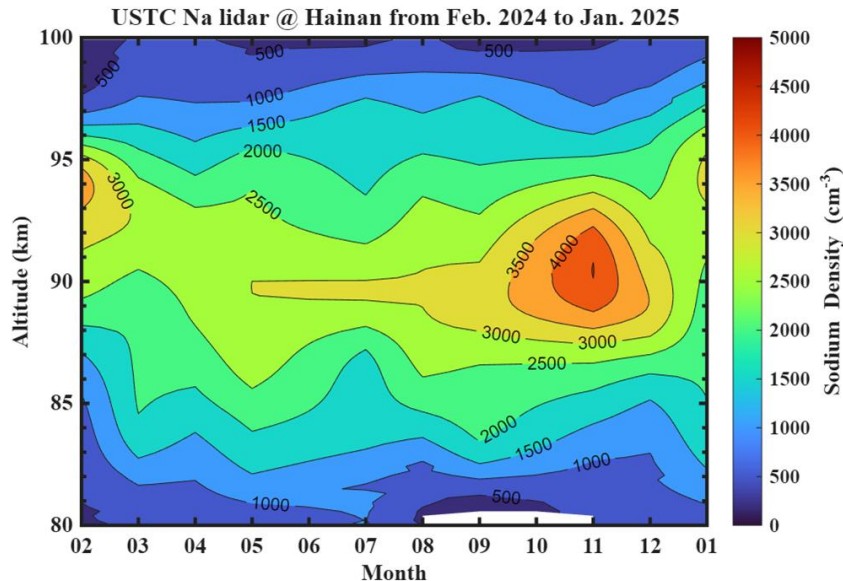

**Figure 9. Monthly mean of nightly sodium density observed by lidar.**

Figure 10 compares monthly mean sodium layer temperatures derived from the Hainan sodium lidar with those from the SABER satellite. For the SABER dataset, points within ±5° latitude/longitude of the lidar site were selected, and only data coinciding with the lidar observation periods were included. These were averaged monthly to represent the temperature distribution.

The vertical temperature structure observed over Hainan (19° N, 109° E) differs slightly from that at Hefei (32° N, 117° E), although monthly means are comparable. Below 95 km, temperatures range from ~185 K to 210 K, while above 95 km they decrease to ~170–185 K (Li et al., 2018).

The seasonal trends of lidar and SABER temperatures over Hainan are generally consistent, though small discrepancies in absolute values remain. Below 95 km, lidar observations report higher autumn–winter temperatures compared with SABER and mid-latitude Hefei measurements. This is likely linked to meridional circulation at the mesopause, with upward motion in the summer hemisphere and downward motion in the winter hemisphere. Above 95 km, lidar temperatures are ~5–10 K lower than SABER, likely due to either the reduced signal-to-noise ratio at higher altitudes in lidar retrievals (Li et al., 2012) or non-local thermodynamic equilibrium (non-LTE) effects in SABER retrievals (Mertens et al., 2001).



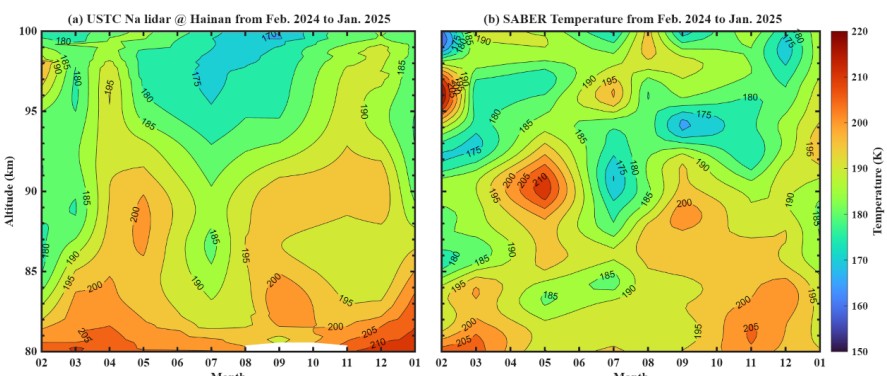

**Figure 10. Monthly mean of nightly mean temperature observed by (a) lidar and (b) SABER.**

## 4. Gravity Wave Vertical Flux

### 4.1 Heat Flux

The vertical fluxes of sensible heat (w′ T′) and sodium density (w′ ρ′$_{Na}$) were calculated using data with vertical and temporal resolutions of $\Delta z$ = 2 km and $\Delta t$ = 30 minutes, respectively. The heat flux over Hainan is shown in Fig. 11a and is predominantly downward, consistent with theoretical expectations (Walterscheid, 1981; Weinstock, 1983). Two distinct downward flux peaks are observed: −0.25 K·m s$^{-1}$ at 83 km and −0.80 K·m s$^{-1}$ at 89 km. By contrast, at Hefei, a single peak of −2 K·m s$^{-1}$ typically occurs

around 88 km except in summer (Li et al., 2022). This difference suggests that the altitude and magnitude of gravity wave dissipation vary across locations.

The corresponding heating rates (Fig. 11b), derived from the divergence of the dynamical heat flux, show a maximum cooling rate exceeding 55 K day$^{-1}$ at 85 km—a value comparable to radiative contributions. These results highlight the critical role of gravity wave dissipation in maintaining the thermal balance of

the mesopause region (Liu and Gardner, 2005).

At the Starfire Optical Range (SOR, 35° N), seasonal variations in sensible heat flux display a strong semiannual pattern, with maximum downward fluxes of −2 to −3 K·m s$^{-1}$ near 88 km during early November to early February, and minima of ∼−0.5 K·m s$^{-1}$ around the equinoxes (Gardner & Liu, 2007). Measurements at Maui, Hawaii (20.7° N), also reveal notable seasonal features. The annual mean heat

flux exhibits a double-peak structure, with two downward maxima of −1.25 K·m s$^{-1}$ at 87 km and −1.4 K·m s$^{-1}$ at 95 km (Liu & Gardner, 2005). This structure is similar to that observed in Hainan, which lies at a comparable latitude but different longitude.





More recently, Guo and Liu (2021) reported seasonal variations in vertical GW heat flux over Cerro Pachón, Chile (30° S). Their results showed strong annual and weaker semiannual oscillations, with maximum downward fluxes in June–July. The flux profile exhibited a broad maximum extending from 88 to 94 km, with average values around −2.5 K·m s⁻¹, consistent with peak values observed in late May at McMurdo.

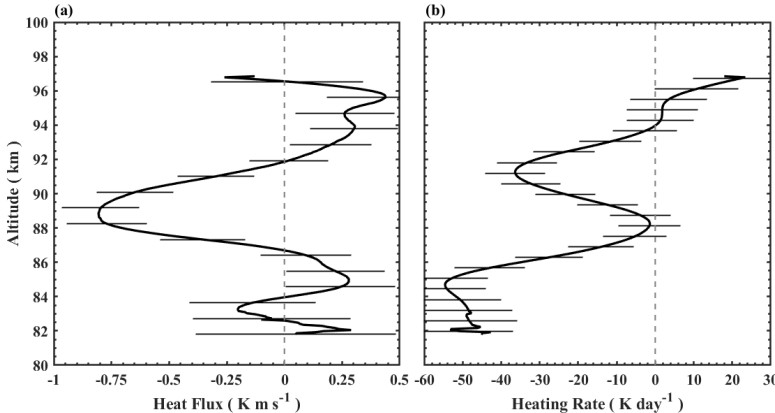

**Figure 11. The vertical flux of sensible heat. (a) Annual mean sensible heat flux derived from vertical wind and temperature measurements at Hainan, China, and (b) the corresponding heating rates. Error bars are shown as thin lines.**

## 4.2 Sodium Flux

The dynamical flux of sodium is shown in Fig. 12a. Over Hainan (19.5° N, 109.1° E), the measured sodium flux exhibits two peaks exceeding −50 m s⁻¹·cm⁻³ at altitudes of 88 and 90 km. This transport results in a net sodium loss peaking near 91 km, with a downward flux rate of ~165 cm⁻³·h⁻¹. At Maui (20.7° N), the maximum flux is slightly larger (−80 m s⁻¹·cm⁻³), while at Hefei (32° N, 117° E), values are smaller (−30 m s⁻¹·cm⁻³) in the 89–95 km altitude range (Chu et al., 2022).

At the Starfire Optical Range (SOR, 35° N), sodium fluxes show strong semiannual variations, with maximum downward values between −175 and −275 m s⁻¹·cm⁻³ near 88 km from early November to early February, and minima around −25 m s⁻¹·cm⁻³ during the equinoxes (Gardner & Liu, 2010). Similarly, observations at Table Mountain, Colorado (40° N), in August–September reported peak values of −150 m s⁻¹·cm⁻³ at 86 km (Huang et al., 2015).

In summary, the heat and sodium fluxes measured at Hainan peak at altitudes similar to those observed at other latitudes and longitudes. However, the absolute magnitudes of sodium fluxes are smaller than at



most other sites. A comparison of annual mean gravity wave heat and sodium fluxes across different

lidar stations is provided in Table 2.

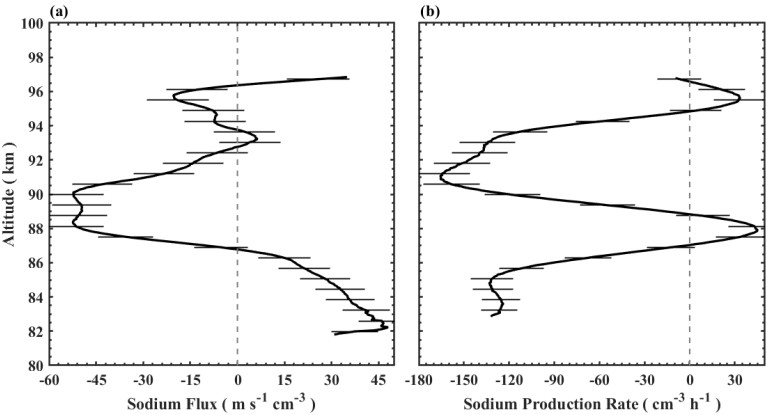

**Figure 12. The dynamical flux of sodium. (a) Na flux due to dissipating gravity waves calculated from the measured Na density and vertical wind measurements at Hainan. (b) Na production rate calculated from**
**measured Na flux at Hainan. Error bars are shown as thin lines.**

**Table 2.** Annual Heat and Na Fluxes at Different Sodium Lidar Stations

| Stations | SOR New Mexico | Hefei, China | Maui, Hawaii | Hainan, China | Cerro Pachón, Chile |
|---|---|---|---|---|---|
| Latitude and longitude | 35.0° N 106.5° W | 31.5° N 117.2° E | 20.7° N 156.3° W | 19.5° N 109.1° E | 30.3° S 70.7° W |
| Resolutions (min)/ (km) | 1.5/0.5 | 10/~3 | 1.5/0.96 | 30/~2 | 1/2 |
| Heat flux peak (K m s⁻¹) | −1.1 | −1.04 | −1.0 | −0.8 | −0.4 |
| Heat flux peak altitude (km) | ~88 | 89–93 | 87–95 | ~89 | ~88 |
| Na flux peak (m s⁻¹·cm⁻³) | / | / | −80 | −50 | / |
| Na flux peak altitude (km) | / | / | 88 | 88–93 | / |

**5. Summary**

The Hainan narrow band sodium lidar, deployed in January 2024 at Hainan, China (19.5° N, 109.1° E), incorporates multiple technical improvements that enable automated operation and enhance usability. A
laser frequency-locking and monitoring module continuously tracks the frequency offset between the transmitted laser and the sodium D2a transition, and its stability is verified through vertical wind



retrievals with a frequency jitter below 1 MHz. This capability significantly improves the accuracy of vertical wind measurements.

Temperature data from the SABER satellite and the MSISE model were used to validate the scientific reliability of the lidar-based temperature retrievals. Monthly variations of sodium density and temperature in the low-latitude region of Hainan were presented. Comparisons with satellite and model data demonstrated generally consistent patterns, with minor discrepancies comparable to those observed at the mid-latitude Hefei site.

The annual mean heat flux over Hainan exhibits two downward maxima: $-0.25$ K·m s$^{-1}$ at 83 km and $-0.80$ K·m s$^{-1}$ at 89 km. Heat flux divergence indicates a net negative heating rate in the mesopause region, contributing approximately $-40$ K day$^{-1}$ to the background atmosphere between 82 and 97 km. Sodium fluxes display two pronounced peaks exceeding $-50$ m s$^{-1}$·cm$^{-3}$ at 88 km and 90 km, with the resulting transport producing a maximum sodium loss rate of 165 cm$^{-3}$·h$^{-1}$ near 91 km. This study therefore provides the first report of seasonal variations in gravity-wave–induced vertical fluxes over Hainan.

Direct measurement of heat flux associated with gravity wave dissipation remains challenging, since the signals are weak relative to the large instantaneous variability of wind and temperature. Substantial temporal averaging is thus required to reduce uncertainties. The present results are derived from more than 300 hours of observations. Continued long-term measurements of wind and temperature with the Hainan sodium lidar will further improve the precision of flux estimates and deepen understanding of gravity-wave–driven processes in the mesopause region.

*Data Availability Statement.* The lidar data and code in this work can be downloaded from Science Data Bank repository at (Wang et al., 2025) https://doi.org/10.57760/sciencedb.27247. The authors also thank SABER team for making the SABER temperature dataset can be available at https://saber.gats-inc.com/browse_data.php.

*Author contributions.* XF and XX conceived the research. XW, XC and WG contributed to the investigation. XW conducted the experiment, characterized the systems, and analyzed the data. WG contributed to the software. XW wrote the manuscript, guided by XF and XX. XF, TL and CY contributed to the scientific discussion. TC, XC and TL provided support for the data curation. TC



contributed to the project administration. XF and XX acquired the research funding. All co-authors
contributed to proofreading of the manuscript.

*Competing interests*. The contact author has declared that none of the authors has any competing interests.

*Acknowledgments*. This work is supported by B-type Strategic Priority Program of the Chinese
Academy of Sciences (XDB0780000), National Natural Science Foundation of China (Grant
No.42394122), the Ground-based Space Environment Monitoring Network (the Chinese Meridian
Project).

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
