# Peer review of "Vertical Distribution of Heat and Sodium Fluxes in the Mesopause Region Measured by Sodium Lidar Over Hainan, China (109°E,19°N)"

_EGUsphere, 2025_

## Referee Comment (RC1)

**Review of egusphere-2025-4803**

The authors describe the design of a new Na Doppler lidar system, and the initial observations of Na density, temperature, and vertical winds made with the instrument over Hainan, China (19  $^{\circ}$  N). The instrument design involves a novel new way to precisely lock the laser to the hyperfine structure of the Na D2 fluorescence, by employing modulation transfer spectroscopy, and by measuring the frequency chirp induced by the optical pulsed dye amplifier. This latter feature enables the measurement of the absolute radial wind. Until now most groups removed the wind bias caused by amplifier chirp by computing and subtracting long-term means, which is effective for vertical wind observations but not for horizontal winds. In the Abstract it is stated "We present the first lidar-based **characterization of seasonal variations in gravity-wave-induced vertical heat flux, sodium flux**, and associated parameters...". This new lidar and its observations at low latitude (19  $^{\circ}$  N) are important additions to our ability to observe the mesopause region and could potentially improve our understanding of the influence of gravity waves in this region. However, the current manuscript is deficient in key areas and should be returned to the authors for major revisions.

Measurements of the vertical fluxes of species, heat, and horizontal momentum are rare so new measurements of the Na and heat fluxes are an important contribution to our knowledge wave-induced transport. Although the authors claim in the Abstract to present seasonal variations of Na and heat fluxes, they only presented what I think are annual mean profiles (Figs. 11 & 12). Furthermore, although their powerful lidar (PA=1.2 Wm²,  $\Delta z$ =61.44 m,  $\Delta t$ =1 min) should be capable of measuring vertical fluxes at high temporal resolution, the data were smoothed to resolutions of  $\Delta z$ =2 km and  $\Delta t$ =30 min, before the computing the vertical fluxes. This is a significant problem, at least for the heat flux, where Guo & Liu (2021) have shown that most of the heat transport is induced by waves with periods less that 1 h, waves which were eliminated from the Hainan dataset by their smoothing procedure. At the very least, the authors need to reprocess their flux data (both heat and Na fluxes) with a temporal resolution of 2.5 or 3 min so that waves with periods of 5-6 min or longer are included. In addition, the authors need to present the seasonal profiles of the Na and heat fluxes as they claimed in the Abstract.

It would be helpful to the scientific community if the authors derive from their monthly mean Na, T, Na flux and heat flux profiles the annual and semi-annual fits and then compare them previous observations. For example, She et al. (2022) https://doi. org/10.1029/2021JD036291 have a nice Figure 1 which shows the annual and semi-annual amplitudes of fits to the T data from Ft. Collins, CO as well as how the mesopause height changes with season. Their Table 3 compares the seasonal temperature variations at

several sites in both hemispheres, including at South Pole. How do the low-latitude Hainan data compare?

Finally, Table 2 in the Hainan paper does not list the Na flux values from SOR and Hefei. Why not? They were clearly provided in the referenced papers. And the data from McMurdo is not included (Chu et al., 2022, <a href="https://doi.org/10.1029/2021JD035728">https://doi.org/10.1029/2021JD035728</a>).

In summary, the authors have developed an important, improved Na Doppler lidar and collected extensive data covering the calendar year, which potentially will be of considerable interest to the upper atmosphere science community. I strongly encourage the authors to consider the revisions that I have suggested. I recommend the paper be returned for major revisions.

---

## Author Comment (AC2)

**Response to reviewer#2**

Thank you for your careful works and valuable comments. The comments and suggestions are very useful for our manuscript. We have carefully studied these comments and suggestions and made some changes in our manuscript.

**Comment 1)**

In line 87, the photons should pass through a collimating lens, a narrow band optical filter and a converging lens, before being focused onto PMT.

**Response:**

Thank you for your comment. In response to feedback, we have revised the description of the optical path. Specifically, the order of components in the sentence has been corrected to: "The photons then pass through a collimating lens, a narrow band optical filter, and a converging lens before being focused onto the active area of a photomultiplier tube (PMT)." on page 4, line 87 of the manuscript.

**Comment 2)**

An improvement of the lidar shown in the manuscript is the frequency offset monitor unit. Was the measured frequency offset corrected in the lidar measurements shown in the Figure 6?

**Response:**

Thank you for your suggestion. Figure 6 displays the uncorrected vertical wind observations: the nightly average (solid line) and 30-minute profiles (dashed lines). The systematic offset in the average, as explained in Section 2.3, is attributed to the characterized laser frequency offset. Therefore, the corrected wind field, which essentially removes this known offset, is not separately plotted.

A comparison in Fig. 1 reveals consistent trends between the nightly-mean-subtracted vertical wind (black dashed line) and the frequency-offset-corrected wind (blue solid line), with the latter resolving finer atmospheric structures. This agreement supports the efficacy of the frequency monitoring module in correcting systematic instrumental

biases.

[Figure]

**Figure 1.** Sodium layer vertical wind (black dashed line) and the vertical wind after correcting frequency offset (blue solid line) at 30 minutes on 7 July 2024. Error bars are shown as thin lines.

**Comment 3)**

In the Figure8, the histogram color of the measurement hours is different from the color in the legend.

**Response:**

Thank you for pointing this out. When adjusting the color scheme of the figures to avoid red–green combinations, we inadvertently neglected to update the corresponding colors in the legends. This inconsistency has now been rectified, as illustrated in Fig. 2, and the same revision will be incorporated into the revised manuscript accordingly.

[Figure]

**Figure 2.** Histogram of number of nights and hours with valid data observed by the USTC sodium lidar at Hainan.